# A Novel Joint Channel Estimation and Symbol Detection Receiver for Orthogonal Time Frequency Space in Vehicular Networks

**DOI:** 10.3390/e25091358

**Published:** 2023-09-20

**Authors:** Xiaoqi Zhang, Haifeng Wen, Ziyu Yan, Weijie Yuan, Jun Wu, Zhongjie Li

**Affiliations:** 1Department of Electronic and Electrical Engineering, Southern University of Science and Technology, Shenzhen 518055, China; 12132160@mail.sustech.edu.cn (X.Z.); lizj32021@mail.sustech.edu.cn (Z.L.); 2Information Hub, The Hong Kong University of Science and Technology (Guangzhou), Guangzhou 511458, China

**Keywords:** integrated sensing and communication (ISAC), orthogonal time frequency space (OTFS), deep learning, message passing (MP), joint channel estimation and symbol detection

## Abstract

A vehicular network embodies a specialized variant of wireless network systems, characterized by its capability to facilitate inter-vehicular communication and connectivity with the encompassing infrastructure. With the rapid development of wireless communication technology, high-speed and reliable communication has become increasingly important in vehicular networks. It has been demonstrated that orthogonal time frequency space (OTFS) modulation proves effective in addressing the challenges posed by high-mobility environments, as it transforms the time-varying channels into the delay-Doppler domain. Motivated by this, in this paper, we focus on the theme of integrated sensing and communication (ISAC)-assisted OTFS receiver design, which aims to perform sensing channel estimation and communication symbol detection. Specifically, the estimation of the sensing channel is accomplished through the utilization of a deep residual denoising network (DRDN), while the communication symbol detection is performed by orthogonal approximate message passing (OAMP) processing. The numerical results demonstrate that the proposed ISAC system exhibits superior performance and robustness compared to traditional methods, with a lower complexity as well. The proposed system has great potential for future applications in wireless communication systems, especially in challenging scenarios with high mobility and interference.

## 1. Introduction

A vehicular network is a type of wireless network that connects vehicles to each other and to the infrastructure around them. It constitutes a paramount technological substrate for intelligent transportation systems (ITS), smart urban environments, and autonomous vehicular navigation systems [1]. Vehicular networks furnish a comprehensive spectrum of onboard data utilities, encompassing enhanced road safety measures, uninterrupted navigational support, sophisticated traffic management, optimized driving comfort, and integrated infotainment provisions [2]. These networks operate as mobile ad hoc networks (MANETs) and employ various technologies such as Wi-Fi, Bluetooth, and dedicated short-range communications (DSRC) to facilitate communication among vehicles, roadside units (RSUs), traffic lights, and other entities. Meanwhile, it has gained extensive usage in providing real-time traffic information, navigation assistance, and entertainment services to drivers and passengers, as well as in monitoring traffic conditions and detecting accidents [3,4]. In addition to efficient communication, vehicular networks also require highly accurate sensing capabilities, typically achieved through radar systems. The ability to accurately sense both the spatial positioning and velocity of vehicles is a critical requirement for effectuating collision avoidance and empowering viable real-time vehicular safety applications. To address this requirement, integrated sensing and communication (ISAC) technology has been applied [5]. By incorporating ISAC technology into the vehicular network, vehicles are capable of simultaneously performing communication and sensing tasks. Furthermore, the sharing of hardware architecture between the sensing and communication systems enabled by ISAC technology significantly reduces hardware deployment costs, a highly advantageous feature in the context of vehicular networks [6].

Recently, orthogonal time frequency space (OTFS) modulation has been gaining attention due to its ability to provide more reliable communications than OFDM, especially in high-mobility environments [7]. Contrary to its OFDM-based ISAC counterpart, the OTFS-aided ISAC system harmoniously integrates the transmission of signals and the respective channel responses within a singular domain for both functionalities, concurrently displaying robustness against delay and Doppler spreads [8]. Moreover, the RSU has the capability to harness the OTFS-ISAC signals to disseminate downlink information to vehicles and concurrently infer their sensor data predicated on the reflective echoes. As such, the OTFS has recently attracted considerable interest as a viable alternative for ISAC systems [5,9,10,11].

As mentioned above, the sensing information of the environment can be obtained from the OTFS channel matrix without additional signaling costs. Accordingly, it is essential for the RSU and users to estimate channel information with low latency accurately. The authors in [12] proposed a message passing algorithm (MPA)-based channel estimation based on the hidden Markov model, which effectively addressed the effects of fractional Doppler for OTFS systems. For the purpose of radar sensing, Ref. [13] proposed a two-dimensional (2D) correlation-based algorithm for OTFS sensing, which overcame the channel estimation in the DD domain and enhanced the estimation accuracy. Furthermore, an off-grid channel estimation method was adopted for estimating effective channel response in the DD domain, which facilitated a reduction in estimation accuracy degradation associated with fractional Doppler. In some intricate scenarios such as underwater acoustic communication with a t-distribution noise, model-based approaches may fall short in addressing channel estimation. Due to the robust data-driven capabilities of deep learning, it is frequently employed to tackle previously intractable issues [14,15,16]. For instance, the authors in [15] employed a deep residual convolution neural network to deal with noise in RIS channels. In [16], the authors leveraged reinforcement learning to address MIMO channel estimation, addressing the computational complexity of algorithms and the time-varying nature of the channel. Inspired by the successful application of deep learning, we employed an innovative neural-network-based technique for OTFS channel estimation.

The rigorous exploitation of the time–frequency diversity facilitated by the wireless channel, associated with the OTFS modulation, generally necessitates a higher level of detection complexity compared to its conventional counterpart, the OFDM modulation. This amplified complexity arises from the distinctiveness of the delay-Doppler (DD) domain channel. In this context, the intercepted signal could be interpreted as an overlay of transmitted signals, each being power-diminished, phase-altered, and subjected to both delay and Doppler shifts, relative to each discernible path of the wireless channel. In order to confront this complexity escalation, a widely accepted solution is the implementation of a message-passing algorithm (MPA) for OTFS detection. Nevertheless, this algorithm may impose significant computational demands. To decrease the detection complexity, a Gaussian approximation technique has been introduced to address the intersymbol interference (ISI) in the DD domain [17]. Recently, a novel extension to the maximum a posterior (MAP), termed the hybrid MAP and parallel interference cancellation (PIC) detection, has been proposed [18]. In this innovative solution, the Gaussian approximation is selectively applied to portions of the DD domain ISI, based on the corresponding path attenuation levels. In a further initiative to streamline receiver complexity, a variational Bayes (VB) methodology was proposed as a surrogate to the optimal MAP detection [19]. Moreover, [20] introduced low-complexity detectors, capable of achieving commendable error rate performance by leveraging the block-diagonal characteristic of the time-domain effective channel matrix, using an SVD-based orthogonal linear estimator.

In this paper, we design a novel joint channel estimation and symbol detection framework for the OTFS-aided ISAC in uplink communications to reduce the signaling overhead, focusing on the dual tasks of channel estimation and symbol detection within vehicular networks. The RSU can leverage the OTFS-ISAC signals to garner estimates of temporal delays, Doppler shifts, and angles related to vehicles encompassed within its communication sphere. Specifically, for channel estimation, a deep neural network (DNN)-based estimator is employed to achieve accurate and robust channel estimation for some complex scenarios where the noise cannot be solved by model-based methods. On the other hand, we adopt a low-complexity algorithm named orthogonal approximate message passing (OAMP) for the symbol detection. To the best of our knowledge, this is the first joint channel estimation and detection scheme based on deep learning designed for uplink communication in OTFS-assisted ISAC scenarios. To further illustrate our work, the contributions are listed as follows:We propose an integrated OTFS-ISAC system that leverages a novel deep residual denoising network and OAMP algorithm for joint channel estimation and symbol detection. Specifically, we design a DNN-based denoising module, incorporating an element-by-element subtraction operation that concurrently exploits the spatial attributes of noise-infected channel matrices as well as the additive character of the perturbation. In addition, a subnetwork that can generate thresholds is utilized to eliminate irrelevant features, thereby enhancing the estimation accuracy.We employ the OAMP detector to carry out the OTFS symbol detection, as it has the potential for MMSE optimality and exhibits excellent detection performance.We demonstrate the effectiveness of the proposed system through simulations and compare its performance with traditional communication systems. The proposed system shows superior performance in challenging environments such as a high Doppler frequency and delay spread, making it a promising solution for future wireless communication systems.

The subsequent sections of this paper are structured as follows. In Section 2, the system model is presented, encompassing the OTFS modulation, the communication model, and the sensing model. In Section 3, the ISAC-based OTFS transmitter design is presented. Section 4 summarizes the experimental results, and Section 5 concludes the paper.

## 2. System Model

As depicted in Figure 1, we consider a classical vehicular network where a roadside unit (RSU) is employed to serve *P* vehicles. In particular, the RSU is equipped with a uniform linear array (ULA) consisting of $N_{t}$ transmit antennas and a separate ULA with $N_{r}$ receive antennas. Assuming there is a sufficient distance between the transmit and receive arrays, the echoes will not interfere with the downlink communication. In addition, the ULAs of the RSU are positioned parallel to the road, resulting in identical angle-of-arrival (AoA) and angle-of-departure (AoD) values for each ULA. We then model the vehicles as point targets and assume that each vehicle is equipped with a single antenna for communicating with the RSU. In the following, we present the signal models of OTFS-assisted ISAC systems.

### 2.1. The Modulation of OTFS Signal

The system diagram of the OTFS modulation is presented in Figure 2. Let us denote *M* and *N* as the number of subcarriers and the number of time slots for each OTFS frame, respectively. The NM information symbols are taken from a modulation alphabet $A=\{a_{1},\cdots ,a_{Q}\}$ comprising *Q* elements. Then, these symbols are arranged in a 2D matrix $X\in C_{N\times M}$ with entries X[k,l], where k∈{0,1,⋯,N−1} and l∈{0,1,⋯,M−1}, respectively. The transmitter proceeds to map the DD domain symbols X[k,l] to NM samples $X_{\mathrm{tf}}$ on the time-frequency grid by using the inverse symplectic fast Fourier transform (ISFFT), which is expressed as
(1)$$\begin{array}{c} X_{\mathrm{tf}}\left[n,m\right]=\frac{1}{MN}\sum_{k=0}^{N-1}\sum_{l=0}^{M-1}X\left[k,l\right]e^{j2\pi \left(\frac{nk}{N}-\frac{ml}{M}\right)}, \end{array}$$
where n∈{0,1,⋯,N−1}, m∈{0,1,⋯,M−1}, and $X_{\mathrm{tf}}$ is the transmitted samples matrix in the time–frequency domain. Based on the TF domain transmitted symbols, we can adopt the conventional OFDM modulator to convert the 2D samples $X_{\mathrm{tf}}$ into a continuous-time waveform x(t) with the aid of a transmit waveform $g_{tx}\left(t\right)$ given by
(2)$$\begin{array}{c} x\left(t\right)=\sum_{n=0}^{N-1}\sum_{m=0}^{M-1}X_{\mathrm{tf}}\left[n,m\right]g_{tx}\left(t-nT\right)e^{j2\pi m\Delta f(t-nT)}, \end{array}$$
where Δf represents the frequency spacing between any adjacent subcarriers. Note that Equation (Equation 2) is the Heisenberg transform in Figure 2, and the Winger transform is the inverse transformation of the Heisenberg transform [21]. By utilizing OTFS signals, communication and sensing can be enhanced more efficiently without requiring additional hardware devices and signal processing.

### 2.2. Communication Signal

During the downlink transmission phase, the RSU can utilize an antenna array for creating either a wide beam or an omnidirectional signal, which enables the detection of all relevant targets within the range. During the tracking mode, the RSU can generate multiple beams to facilitate information transmission and target tracking. The multibeam signal with a beamforming matrix $W\in C^{N_{t}\times P}$ can be expressed as
(3)$$\begin{array}{c} \tilde{x}\left(t\right)=Wx\left(t\right), \end{array}$$
where $x\left(t\right)=[x_{1}\left(t\right),\cdots ,x_{P}\left(t\right)]$ is the transmitted signal to all targets. The *i*th column of W is denoted as $w_{i}=\sqrt{\frac{p_{i}}{N_{t}}}aN_{t}\left(\tilde{\theta }i\right)$, fulfilling both power allocation and directional steering functions. Specifically, the term $p_{i}$ represents the power allocation factor for the *i*th column of W, and the column vector $a_{N_{t}}\left({\tilde{\theta }}_{i}\right)$ represents the steering vector pointing to the desired direction $\theta_{i}$, which is given by
(4)$$\begin{array}{c} a_{N_{t}}\left(\theta_{i}\right)={\left[1,e^{j\pi \sin\theta_{i}},\ldots ,e^{j\left(N_{t}-1\right)\pi \sin\theta_{i}}\right]}^{T}. \end{array}$$
Thanks to the asymptotic orthogonality of the massive antenna array, the *i*th target’s communication channel is predominantly line-of-sight (LoS) after transmitting the beamformer, which is expressed as
(5)$$\begin{array}{c} C_{\mathrm{DD}}^{i}\left(\tau ,\nu \right)=h_{i}a_{N_{u}}\left(\theta_{i}\right)a_{N_{t}}^{H}\left(\theta_{i}\right)\delta \left(\tau -\tau_{i}\right)\delta \left(\nu -\nu_{i}\right), \end{array}$$
where $h_{i}=\sqrt{\frac{c}{4\pi f_{c}d_{i}^{2}}}$ represents the channel gain. Here, the term *c* represents the signal propagation speed, $f_{c}$ is the carrier frequency, $d_{i}$ is the range distance to the *i*th target, and $\tau_{i}$ and $\nu_{i}$ represent the delay and Doppler shift, respectively. The received signal can be written as
(6)$$\begin{array}{c} y_{i}\left(t\right)=h_{i}u_{i}^{H}a_{N_{u}}\left(\theta_{i}\right)a_{N_{t}}^{H}\left(\theta_{i}\right)w_{i}x_{i}\left(t-\tau_{i}\right)e^{j2\pi \nu_{i}\left(t-\tau_{i}\right)}+z_{i}\left(t\right), \end{array}$$
where $u_{i}\in C^{N_{u}\times 1}$ represents the received beamformer, and $z_{i}\left(t\right)$ denotes the noise signal in the time domain. With a received pulse shaping filter $g_{rx}\left(t\right)$, the expression for the received OTFS signal in the DD domain can be obtained by the Winger transform and the symplectic Fourier transform (SFFT), i.e.,
(7)$$\begin{array}{c} Y\left[l,k\right]=h_{i}ui^{H}aN_{u}\left(\theta_{i}\right)a_{N_{t}}^{H}\left(\theta_{i}\right)f_{i}X\left[{\left(l-l_{i}\right)}_{M},{\left(k-k_{i}\right)}_{N}\right]+Z\left[l,k\right], \end{array}$$
where the integers $k_{i}=\nu_{i}NT$ and $l_{i}=\tau_{i}M\Delta f$, and Z[l,k] denotes the independent white Gaussian noise sample with a power spectral density of $N_{0}$.

### 2.3. Sensing Signal

In the downlink transmission, the information about the environment can be acquired from the OTFS channel matrix, which is given by
(8)$$\begin{array}{c} H\left(t,\tau \right)=\sum_{i=1}^{K}\gamma_{i}a_{N_{r}}\left(\theta_{i}\right)a_{N_{t}}^{H}\left(\theta_{i}\right)\delta \left(\tau -\eta_{i}\right)e^{j2\pi v_{i}t}, \end{array}$$
where $\gamma_{i}$, $\eta_{i}$, and $\nu_{i}$ represent the reflected coefficient, delay, and Doppler shift corresponding to the *i*th target, respectively. The received sensing echoes at the RSU can then be expressed as:(9)$$\begin{array}{c} r\left(t\right)=\sum_{i=1}^{K}\gamma_{i}a_{N_{r}}\left(\theta_{i}\right)a_{N_{t}}^{H}\left(\theta_{i}\right)\tilde{x}\left(t-\eta_{i}\right)e^{j2\pi v_{i}t}+n\left(t\right), \end{array}$$
where n(t) denotes the additional measurement noise. Recall that the steering vectors with different angular values are asymptotically orthogonal for massive MIMO receive antenna arrays [8], i.e., $a_{N_{r}}^{H}\left(\theta_{i}\right)a_{N_{r}}\left(\theta_{i0}\right)\approx 0$ for $\theta_{i}\neq \theta_{i0}$. This implies that we can omit the interference resulted from different targets in the sensing echoes and the RSU is capable of differentiating various targets based on their angles of arrival (AoAs). Therefore, one can extract the sensing echo from the *i*th target from r(t) using a receive beamformer $b_{i}=a_{N_{r}}^{H}\left({\tilde{\theta }}_{i}\right)$, which can be written as:(10)$$\begin{array}{c} r_{i}\left(t\right)=\gamma_{i}b_{i}^{H}a_{N_{r}}\left(\theta_{i}\right)a_{N_{t}}^{H}\left(\theta_{i}\right)f_{i}s_{i}\left(t-\eta_{i}\right)e^{j2\pi v_{i}t}+n\left(t\right). \end{array}$$
It is noteworthy that the angular parameter $\theta_{i}$ can also be inferred when processing the receive beamformer by comparing the gains obtained from different beam directions. Therefore, Equation (Equation 9) can be recast as:(11)$$\begin{array}{c} r_{i}\left(t\right)=G_{a}x_{i}\left(t-\eta_{i}\right)e^{j2\pi v_{i}t}+n_{i}\left(t\right), \end{array}$$
where $G_{a}$ denotes the composite antenna array gain. As a step forward, by means of an ideal receive filter and operating OTFS demodulation, we have the input–output relationship in the DD domain given by:(12)$$\begin{array}{c} R\left[l,k\right]=\;G_{a}\sum_{k^{\prime }=0}^{N-1}\sum_{l^{\prime }=0}^{M-1}H\left[l^{\prime },k^{\prime }\right]\cdot X\left[{\left(l-l^{\prime }\right)}_{M},{\left(k-k^{\prime }\right)}_{N}\right]+n\left[l,k\right],\; \end{array}$$
where $H[l^{\prime },k^{\prime }]$ represents the gain of the *i*th target at the DD grid (bin) with indices $l_{0}$ and $k_{0}$ corresponding to the delay of $\frac{l^{\prime }}{M\Delta f}$ and Doppler of $\frac{k^{\prime }}{NT}$.

### 2.4. JCESD for OTFS-Based Vehicular Networks

In this subsection, we first present the OTFS-based ISAC system framework as illustrated in Figure 3. At a time slot, the users receive the y(t) from the RSU via beam assignment, while the RSU receives the reflected signal r(t) from the vehicles. Given the received echo, the RSU performs OTFS demodulation to obtain the DD domain signal R(l,k) as that establishes high-quality communication service quality, and the RSU needs to obtain information about the vehicles’ azimuth angles. As mentioned before, the vehicle status information can be acquired from the outcomes of the OTFS channel estimation $C_{DD}\left(\tau ,\nu \right)$, thus the RSU is capable of utilizing its antenna arrays to create “pencil-like” beams, enabling a precise alignment with user locations. On the other hand, based on the received signal, users estimate the channel matrix, containing sensing information about the surrounding environment. More importantly, the receiver must accurately and efficiently demodulate symbol information based on the channel information. For this purpose, a low-complexity OAMP detection technique is employed.

## 3. The Joint Channel Estimation and Symbol Detection

To ensure precise CSI for sensing and communication, we introduce a deep learning (DL)-based framework for the OTFS-assisted ISAC system, as illustrated in Figure 4. In this section, we first present an embedded pilot-aided scheme, which involves incorporating pilot symbols into the transmitted symbol matrix to facilitate the preprocessing of the channel estimation. Next, we convert the OTFS channel estimation problem into an issue about sparse image denoising. In particular, we utilize a DL-based framework for accurate channel estimation. Finally, upon acquiring the precise OTFS channel matrix, the RSU performs the OAMP detection to demodulate symbols effectively.

### 3.1. Pilot Placement

Pilot symbols can be inserted into the transmitted signal to aid the channel estimation, which is applicable in the OTFS-assisted ISAC system, as shown in Figure 5. In this paper, we consider the case of integer delay and Doppler frequencies. Let $x_{p}=X_{dd}\left[l_{p},k_{p}\right]$ represent the pilot symbol.

The entries of the DD domain matrix satisfy [22]
(13)$$X_{dd}\left[l,k\right]=\left\{\begin{array}{cc} x_{p} & l=l_{p},k=k_{p},\\ 0\; & l\in [l_{p}-l_{max},l_{p}+l_{max}],\\ & k\in [k_{p}-2k_{max},k_{p}+2k_{max}],\\ \mathrm{data}\;\mathrm{symbol} & \mathrm{otherwise}.\; \end{array}\right.,$$
where $l_{max}$ and $k_{max}$ denote the spacing between the data symbol and the pilot symbol, which also corresponds to the maximum delay and Doppler shift of the target. Define $X_{d}$ and $X_{p}$ as the data matrix and pilot matrix which satisfy
(14)$$X_{d}\left[l,k\right]=\left\{\begin{array}{cc} 0 & l\in [l_{p}-l_{max},l_{p}+l_{max}]\\ & k\in [k_{p}-2k_{max},k_{p}+2k_{max}],\\ \mathrm{data}\;\mathrm{symbol} & \mathrm{otherwise}.\; \end{array}\right.$$
and
(15)$$X_{p}\left[l,k\right]=\left\{\begin{array}{cc} x_{p} & l=l_{p},k=k_{p},\\ 0 & \mathrm{otherwise}.\; \end{array}\right..$$
Subsequently, Equation (Equation 7) can be reformulated as
(16)$$\begin{array}{cc} Y_{dd}\left[l,k\right] & =H_{dd}\left[l,k\right]⊛X_{dd}\left[l,k\right]+Z_{dd}\left[l,k\right]\\ & =H_{dd}\left[l,k\right]⊛\left(X_{d}\left[l,k\right]+X_{p}\left[l,k\right]\right)+Z_{dd}\left[l,k\right], \end{array}$$
where ⊛ denotes the cyclic convolution operation. In order to obtain the CSI by using the pilot symbol, we introduce a received pilot matrix $Y_{p}\in C^{M\times N}$, which satisfies
(17)$$Y_{p}\left[l,k\right]=H_{dd}\left[l,k\right]⊛X_{p}\left[l,k\right]+Z_{dd}\left[l,k\right].$$
Since $X_{p}$ is known by the RSU, and $Y_{p}$ can be obtained by setting some entries of $Y_{dd}$ to 0, the channel estimation problem in the proposed framework can be formulated as a sparse image denoising problem, represented by:(18)$$Y_{p}=H_{p}+Z_{dd},$$
where
(19)$$H_{p}\left[l,k\right]=H_{dd}\left[l,k\right]⊛X_{p}\left[l,k\right].$$
Here, the matrix $H_{p}\left[l,k\right]$ incorporates the OTFS effective channel $H_{dd}\left[l,k\right]$, where each element is characterized by a Bernoulli–Gaussian distribution instead of a Gaussian PDF. Consequently, employing a Bayesian general linear model is unfeasible for the data model, e.g., a linear minimum mean square error (LMMSE). Moreover, the noise component $Z_{dd}$ is not limited to a Gaussian distribution but can include arbitrary noise. As a result, obtaining an explicit expression for a Bayesian estimator within the model-driven approach becomes intractable. In contrast, we employ a data-driven approach to introduce convolutional neural networks (CNNs)-based channel estimation framework for OTFS systems in the subsequent section.

### 3.2. The Architecture of the DL Network

Note that CNNs possess a favorable capability to extract features from matrices containing noisy observations. In addition, the subtraction architecture employed in deep neural networks (DNNs) facilitates the exploration of the additive characteristics of the noise [23,24]. Based on the CNNs and residual subtraction architecture [15], we developed a deep residual denoising network (DRDN) to effectively learn and eliminate the residual noise from received symbols while eliminating irrelevant features. Figure 6 illustrates the general structure of the DRDN. It comprises several components, including an input transformation layer, *D* denoising blocks, and an adaptive soft threshold layer. The hyperparameters associated with the DRDN are presented in Table 1. In the subsequent sections, we provide an introduction to each layer of the network.
(20)$$Y=\left[\begin{array}{c} \left[{Y_{p}}_{ℜ}\right]\\ \left[{Y_{p}}_{ℑ}\right] \end{array}\right],H=\left[\begin{array}{c} \left[{H_{p}}_{ℜ}\right]\\ \left[{H_{p}}_{ℑ}\right] \end{array}\right],$$
where $Y\in C^{2k_{\max}\times l_{\max}\times 2}$ and $H\in C^{M\times N\times 2}$ are the real-valued received signals and the channel coefficients, respectively. Here, subscripts *ℜ* and *ℑ* represent the real and imaginary components of the input variable, respectively. Therefore, it is possible to devise an efficient denoiser using a DRDN approach to enhance the OTFS channel estimation.

Denoising Block: Convolutional Neural Networks (CNNs) are widely employed in numerous DL applications [25,26] due to their remarkable feature extraction capabilities. As illustrated in Figure 6, the proposed DRDN model follows a general architecture. The denoising performance is gradually enhanced by leveraging the *D* denoising blocks with identical structures. Each denoising block comprises a residual subnetwork and an elementwise subtraction operation. The residual subnetwork consists of *L* layers that employ three types of functions. In the first layer, the “Conv+BN+ReLU” composition is employed to extract the spatial characteristics of the channel matrix. This composition involves applying convolution (Conv) and rectified linear unit (ReLU) operations. Batch normalization (BN) is incorporated between the Conv and ReLU operations to enhance network stability and expedite training. The second composition, with L−2 layers, employs the same structure but employs a kernel size suitable for nonlinear transformations among the features. The ultimate layer employs a solitary convolution operation to generate the noise matrix for subsequent elementwise subtraction. Specifically, the denoised channel matrix is obtained through the process of subtracting each element of the residual subnetwork from the corresponding element of the original input.
(21)$$T_{N}=I-\sum_{i=0}^{D-1}S_{i}=I-\sum_{i=0}^{D-1}f_{\theta_{i}}\left(I_{i}\right).$$
Here, $f_{\theta_{i}}$, $T_{N}$, and I denote the function of the *i*th DB, the output, and input of DB. The residual term, referred to as the residual noise, is denoted as $S_{i}$.

DL-based threshold layer: At the last layer, a soft shrinkage function is designed to represent the sparse features, which is inspired by sparse code in image denoising [27]. To choose an appropriate threshold, we designed an adaptive thresholding layer, where the threshold is determined by a subnetwork. Figure 7 illustrates the proposed adaptive thresholding module, which begins by applying a nonlinear operation to the feature map $T_{N}$ obtained from the denoising module. The subnet $f_{1}\left(\cdot \right)$ is adopted to obtain the coarse estimation $\lambda_{t}$. In particular, the global average pooling (GAP) is used to acquire C×1×1 dimensional output. Subsequently, $\lambda_{t}$ is propagated through a two-layer fully connected network to calculate the scaling parameter α using the sigmoid function, i.e., $\alpha =\mathrm{sigmoid}(\lambda_{t})$. This scaling parameter α represents the threshold scaling factor, which is used to compute the final threshold given by
(22)$$\begin{array}{c} \lambda =\alpha \lambda_{t} \end{array}$$
The final channel estimation is performed by comparing each element in $T_{N}$ with the threshold λ. If an element is greater than λ, it is retained; otherwise, it is set to 0, effectively performing a filtering operation. Mathematically, it can be expressed as follows
(23)$$\begin{array}{c} \eta \left(\lambda ,T_{N}\right)=\mathrm{sign}\left(T_{N}\right)\max\{|T_{N}\left|-\lambda ,0\right\}, \end{array}$$
where η is the adaptive threshold filtering function, sign(·) denotes the sign function, and its value is 1 when the value of x is greater than zero and −1 when it is less than 0.

Output layer: At the end of the neural network, the effective area $\tilde{Y_{p}}\in C^{2k_{\max}\times l_{\max}}$ needs to be converted to complete channel matrix $H_{dd}\in C^{M\times N}$, and is then applied to the sensing and communication. In summary, this paper introduces the DRDN architecture in channel estimation to improve the denoising performance. The architecture incorporates *D* denoising blocks, which systematically eliminate complex noise in unmodeled scenarios. Additionally, a DL-based soft shrinkage operation is employed to accurately determine the delay and Doppler parameters, which in turn facilitate effective target sensing and OTFS symbol detection.

### 3.3. Estimation of Neural Network

Utilizing the DRDN architecture as a foundation, we subsequently develop a channel estimation scheme which comprises two distinct phases: offline training and online estimation.

Offline training: In the offline training phase, a large quantity of data are used to train the network, eventually resulting in a trained model. The training data are denoted by
(24)$$\begin{array}{c} \left(\tilde{Y},H\right)=\left\{\left({\tilde{Y}}^{\left(1\right)},H^{\left(1\right)}\right),\cdots ,\left({\tilde{Y}}^{\left(N\right)},H^{\left(N\right)}\right)\right\}, \end{array}$$
where $\left({\tilde{Y}}^{\left(i\right)},H^{\left(i\right)}\right)$, i∈{1,2,…N}, is the *i*th training example of $\left(\tilde{Y},H\right)$. Moreover, ${\tilde{Y}}^{\left(i\right)}\in C^{2k_{\max}\times l_{\max}}$, $H^{\left(i\right)}\in C^{2k_{\max}\times l_{\max}}$ are the input of the DRDN and the label. According to the mean square error (MSE) criterion, the cost function of the offline training phase can be formulated as
(25)$$J_{MSE}\left(\theta \right)=\frac{1}{N}\sum_{i=1}^{N}{∥f_{\theta }\left(\tilde{Y_{i}}\right)-H_{i}∥}_{2}.$$
Based on this formulation, the DRDN model can utilize the backpropagation (BP) algorithm to achieve effective training. Specifically, the loss function is equivalent to the MMSE estimator when N→∞ [25]. That is, the performance of the DRDN estimator tends to converge to the optimal estimator as the size of the training data increases sufficiently.

Online estimation: In the online estimation, the initial coarse channel estimation result, denoted as ${\tilde{Y}}_{test}$, is obtained from the received symbols. Subsequently, the data are sent into the DRDN estimator, and the process of online estimation can be represented as follows:(26)$$H_{est}=h_{\theta }\left(f_{\theta }\left({\tilde{Y}}_{test}\right)\right),$$
where $h_{\theta }$ denotes the hard-shrink operation. The developed channel estimation method based on the DRDN is a universal approach that enhances system performance by achieving superior channel estimation.

### 3.4. Communication Symbol Detection

In this section, we provide a brief introduction to the OAMP algorithm that we utilized to perform the OTFS symbol detection in the ISAC system, where the estimated effective channel matrix from the DRDN is utilized.

Now, let us develop the OTFS detection problem within the perspective of OAMP. To the best of our knowledge, the OTFS system can be considered as a linear system under a linear constraint Γ as well as a nonlinear constraint Φ, given by:
(27a)$$\begin{array}{cc} \mathrm{Linear}\;\mathrm{constraint}\;\Gamma :\; & y={\hat{H}}^{\mathrm{eff}}x+\tilde{n}, \end{array}$$
(27b)$$\begin{array}{cc} \mathrm{Nonlinear}\;\mathrm{constraint}\;\Phi :\; & X[m,n]\in A,\forall m,n, \end{array}$$
where ${\hat{H}}^{\mathrm{eff}}\in C^{MN\times MN}$ is the estimated effective channel matrix, $\tilde{n}$ represents the additive white Gaussian noise with half power spectral density $N_{0}$, and A is the constellation alphabet (e.g., QPSK) with cardinality |A|. Our objective is to find the minimum mean square error (MMSE) estimation of x, i.e.,
(28)$$\hat{x}=E\left\{x|y,{\hat{H}}^{\mathrm{eff}},\Gamma ,\Phi \right\}.$$
The aforementioned problem can be solved by OAMP with the aid of an iterative method that includes an orthogonal linear estimator (LE) γ and an orthogonal nonlinear estimator (NLE) ϕ. We now invoke the OAMP following the standard procedures shown in Ma et al. [28] as:
(29a)$$\begin{array}{cc} \mathrm{Orthogonal}\;\mathrm{LE}\;\gamma :\; & {\hat{x}}^{\gamma \to \phi }=\gamma \left({\bar{x}}^{\phi \to \gamma }\right), \end{array}$$
(29b)$$\begin{array}{cc} \mathrm{Orthogonal}\;\mathrm{NLE}\;\phi :\; & {\bar{x}}^{\phi \to \gamma }=\phi \left({\hat{x}}^{\gamma \to \phi }\right). \end{array}$$

The core of OAMP is that both the LE and NLE should be designed orthogonally [28]. Hence, the correlation problem, arising from the errors of the iterative process, can be solved thanks to the orthogonality of LE and NLE. Furthermore, the orthogonality can also make the iterative process gradually and steadily converge to the MMSE. It should be noted that both the orthogonality LE and NLE hold if and only if the following constraints are satisfied for iteration l≥0:(30)$$\begin{array}{cc} E\left\{{\left(\xi_{l}^{\gamma \to \phi }\right)}^{H}\xi_{l}^{\phi \to \gamma }\right\} & =0,\\ E\left\{{\left(\xi_{l}^{\phi \to \gamma }\right)}^{H}\xi_{l+1}^{\gamma \to \phi }\right\} & =0, \end{array}$$
where $\xi_{l}^{\gamma \to \phi }$ and $\xi_{l}^{\phi \to \gamma }$ represent the associated Gram–Schmidt (GS) errors, which are written as
(31a)$$\begin{array}{cc} {\hat{x}}_{l}^{\gamma \to \phi } & =\alpha_{l}^{\gamma \to \phi }x+\xi_{l}^{\gamma \to \phi }, \end{array}$$
(31b)$$\begin{array}{cc} {\bar{x}}_{l}^{\phi \to \gamma } & =\alpha_{l}^{\phi \to \gamma }x+\xi_{l}^{\phi \to \gamma }, \end{array}$$
with $\alpha =\frac{1}{N}E\left\{{\hat{x}}^{T}x\right\}$ and $E\{x^{T}\xi \}=0$. The optimal OAMP is then constructed for both LE and NLE using the GS orthogonalization (GSO) given by: (32)$$\begin{array}{cc} \mathrm{Orthogonal}\;\mathrm{LE}\;\gamma :\;{\hat{x}}_{l}^{\gamma \to \phi } & =\gamma ({\bar{x}}_{l}^{\phi \to \gamma })\\ \; & =\hat{\gamma }\left({\bar{x}}_{l}^{\phi \to \gamma }\right)-B_{\hat{\gamma }}{\bar{x}}_{l}^{\phi \to \gamma },\\ \mathrm{Orthogonal}\;\mathrm{NLE}\;\phi :\;{\bar{x}}_{l+1}^{\phi \to \gamma } & =\phi ({\hat{x}}_{l}^{\gamma \to \phi })\\ \; & =\hat{\phi }\left({\hat{x}}_{l}^{\gamma \to \phi }\right)-B_{\hat{\phi }}{\hat{x}}_{l}^{\gamma \to \phi }, \end{array}$$
where $\hat{\gamma }$ is a linear MMSE (LMMSE) estimator, $\hat{\phi }$ represents an MMSE detector, e.g., a symbolwise constellation demapper, $B_{\hat{\gamma }}$ and $B_{\hat{\phi }}$ denote the respective generalized signal orthogonalization (GSO) coefficients. These coefficients are designed to fulfill the orthogonality conditions stated in Equation (Equation 30). For instance, the value of $B_{\hat{\gamma }}$ for the LMMSE estimator is expressed as
(33)$$B_{\hat{\gamma }}=v_{\hat{\gamma }}/v_{\bar{x}},$$
where $v_{\hat{\gamma }}=\frac{1}{N}E\{||\hat{\gamma }\left({\bar{x}}_{l}^{\phi \to \gamma }\right)-x{||}^{2}\}$ and $v_{\bar{x}}=\frac{1}{N}E\{||\xi_{l}^{\phi \to \gamma }{||}^{2}\}$. The computational complexity of OAMP consists of the complexity of the orthogonal LE and orthogonal NLE, where the orthogonal LE contains the LMMSE and orthogonalization operations, whose complexities are $O({\left(MN\right)}^{3})$ and O(MN), respectively, and the nonorthogonal NLE mainly contains the constellation demapper and orthogonalization operations, both of which have a complexity of O(MN). Hence, the overall complexity of OAMP is $O({\left(MN\right)}^{3}+MN)$.

## 4. Simulation Result

In this section, we substantiate the efficiency of the proposed algorithm based on the DRDN through a comprehensive analysis of simulation outcomes. We also provide details of the simulation setups.

### 4.1. Simulation Setups

In this study, the OTFS frame was configured with the parameters *N* = 32 and *M* = 32, indicating the presence of 32 time slots and 32 subcarriers in the time–frequency (TF) domain, as [3]. According to [29], the carrier frequency can be set to 3 GHz, while the subcarrier spacing was maintained at 7.5 kHz. To characterize the channel, we considered a total number of paths denoted as *P*, which was set to six. Due to the limitation of the size of the OTFS data frame, the maximum Doppler shift and maximum delay shift were normalized and denoted by $k_{\max}=4$ and $l_{\max}=5$, respectively. The channel gain was modeled by a complex Gaussian distribution with zero mean and a variance of $\frac{1}{P}$ [30]. Finally, we employed quadrature amplitude modulation (QAM) for efficient bit mapping in our system. For the neural network training, we used the Monte Carlo method to generate $2\times 10^{4}$ samples. The learning rate was set as 0.01, the batch size was B=256, and a weight decay = 0.001 was adopted to deal with overfitting.

### 4.2. Sensing Channel Estimation

To evaluate the performance of the channel estimation, a comparison of the proposed DL-based method with four baseline algorithms (i.e., the LS, the LMMSE, the orthogonal match pursuit (OMP) [31], and the threshold-based method) was made. Note that the threshold-based method is the optimal algorithm in scenarios with a Gaussian white noise when the noise variance is known at the receiver [22]. In addition, the evaluation metric employed was the normalized mean squared error (NMSE), which is defined as $\mathrm{NMSE}=10\log_{10}\frac{||\hat{H}{-H||}_{2}^{2}}{{||H||}_{2}^{2}}$, where $\hat{H}$ and H are the estimated and ground truths, respectively.

As shown in Figure 8, we assessed the proposed methods in scenarios involving a correlated noise [32] and a noise following a t-distribution [23], which are prevalent in real-world settings. It can be observed that a performance gap exists between the LS estimator and the LMMSE estimator, as seen in Figure 8a. This disparity arises because the LS estimator treats the channel as an unknown constant with deterministically defined attributes. In contrast, the LMMSE estimator involves the computation of the noise covariance matrix, leading to a performance improvement of approximately 3 dB compared to the LS method in correlated-noise scenarios. The OMP method frequently employed in compressed sensing algorithms outperforms both the LS and LMMSE algorithms, owing to its consideration of the inherent sparsity within the recovery channel. Nevertheless, the performance of this iterative algorithm experiences a significant degradation due to the stringent convergence conditions and the influence of noise. In comparison to the previous methods, the threshold-based methods perform better when the noise covariance is known at the receiver. Under sparse channel conditions, most errors come from grid points other than the main path. By setting an appropriate threshold, these noises can be effectively removed, resulting in a better performance and robustness compared to the LS and LMMSE methods. In comparison to the traditional LS, LMMSE, and threshold-based methods, deep learning neural networks can more accurately estimate channel characteristics and have a better robustness. By incorporating training into the network design, the absolute value of the residual information can weaken the impact of the noise to a certain extent. The proposed deep learning neural network outperforms existing threshold-based and linear detection methods, improving the performance by approximately 8 dB compared to the OMP method and approximately 2 dB compared to the threshold-based method. As depicted in Figure 8a, in the scenario of a t-distribution noise, it is noticed that the proposed DRDN demonstrates a significant enhancement in estimation performance, outperforming all other considered algorithms. This is due to the inability of the model-based approach to incorporate prior knowledge from the data. In contrast, the DRDN can substantially enhance channel estimation performance due to its robust data-driven capability and the incorporation of sparse prior information.

Figure 9 presents comparisons of the NMSE and BER achieved by the proposed joint channel estimation and symbol detection scheme, as well as by several benchmarks, including the LS, LMMSE, MP, and threshold-based methods. As illustrated in Figure 9, both threshold-based methods and DNN-based methods have superior channel estimation performance compared to traditional LS and LMMSE methods. Note that the LMMSE-based method considers the variance of the noise, resulting in a performance approximately 5 dB higher than the LS method. In comparison with the threshold-based and LMMSE methods, the DRDN method performs better, with a performance approximately 7 dB higher than the OMP algorithm. In the Gaussian noise scenario, the threshold method is the optimal solution when the noise level is ideally acquired at the receiver. In practice, the DRDN has a higher adaptability and generalization ability in complex noise scenarios. The DRDN can accurately complete noise reduction tasks, effectively distinguishing the main channel, and restore the channel.

On the other hand, the results show that when the estimated channel information from the DRDN is used as input, the proposed scheme achieves superior performance compared to the other detectors. Specifically, the OAMP detector exhibits the best performance, while the performance of the MP detector is close to that of OAMP. Furthermore, the results in Figure 9 demonstrate the robustness of OAMP, which maintains good performance even at lower SNR ($E_{b}/N_{0}$) and with imperfect channel estimation. Overall, these findings confirm the effectiveness of the proposed joint channel estimation and symbol detection scheme.

## 5. Conclusions

In this paper, we focused on the theme of ISAC-assisted OTFS receiver design, which aims to perform channel estimation for sensing purposes and detect communication symbols. In order to achieve an effective estimation of the channel, we proposed the utilization of a novel DRDN within our framework. The DRDN was meticulously designed, incorporating a denoising block based on CNNs that encompassed an elementwise subtraction structure. This unique architectural feature enabled the network to exploit both the spatial characteristics of noisy channel matrices and the inherent additive properties of the noise simultaneously. By capitalizing on the robust feature extraction and denoising capabilities of CNNs, our proposed DRDN method showcased an exceptional accuracy in channel estimation, surpassing alternative approaches in the field. The CNN-based channel estimation approach for OTFS had a good potential with promising directions for future work, including solving the fractional delay-Doppler cases and a novel network structure that may require a smaller number of trainable parameters. For the OTFS symbol detection, we utilized the OAMP detector, which had the potential for MMSE optimality and exhibited excellent detection performance. Through simulations, we demonstrated the effectiveness of the proposed system and compared its performance with traditional communication systems. The proposed system exhibited superior performance in challenging environments such as a high Doppler frequency and delay spread, making it a promising solution for future wireless communication systems.

## Figures and Tables

**Figure 1 entropy-25-01358-f001:**
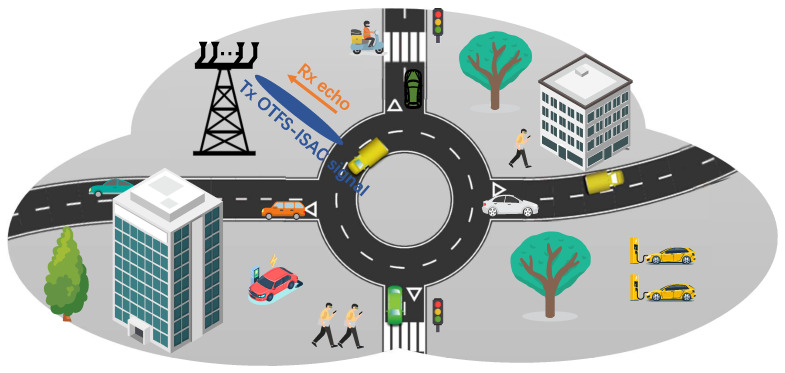
The ISAC-OTFS vehicular network.

**Figure 2 entropy-25-01358-f002:**
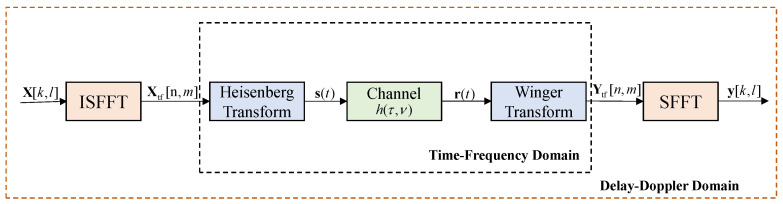
OTFS modulation.

**Figure 3 entropy-25-01358-f003:**
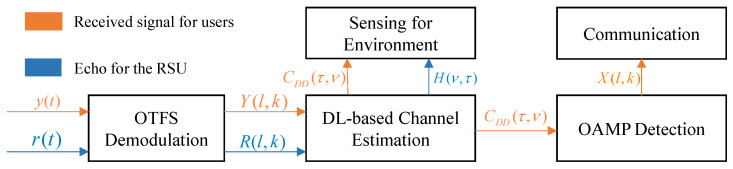
JCESD for OTFS-Based vehicular networks.

**Figure 4 entropy-25-01358-f004:**
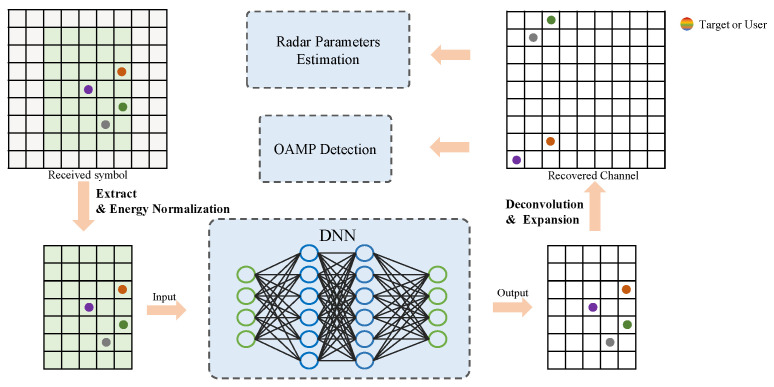
The framework for JCESD.

**Figure 5 entropy-25-01358-f005:**
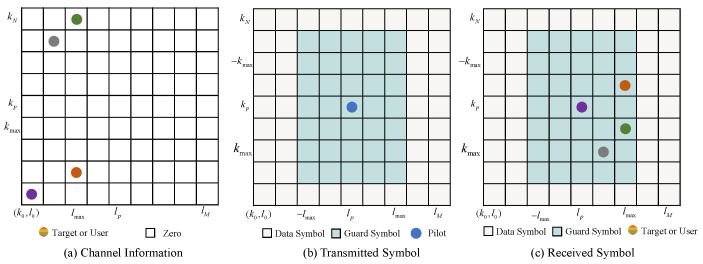
The embedded pilot scheme.

**Figure 6 entropy-25-01358-f006:**
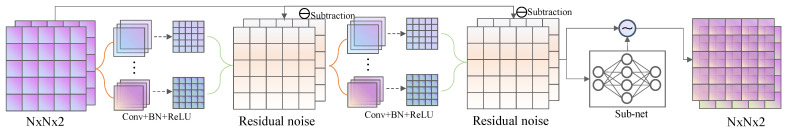
Architecture of DRDN.

**Figure 7 entropy-25-01358-f007:**
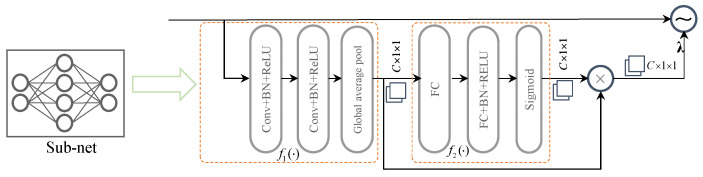
The subnet for the adaptive threshold.

**Figure 8 entropy-25-01358-f008:**
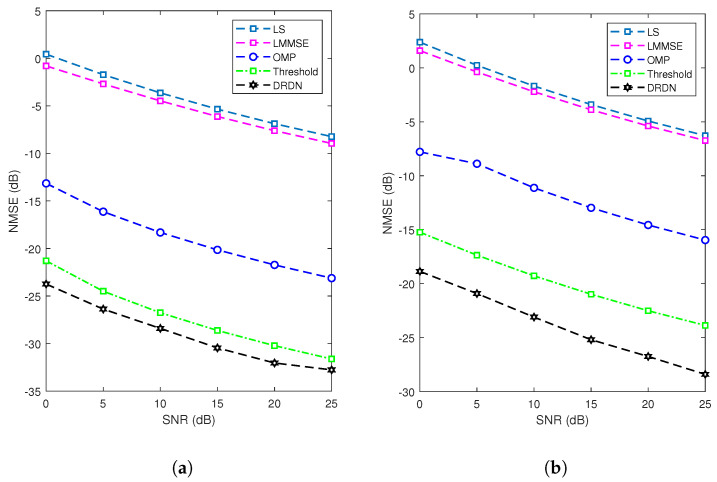
NMSE comparisons with the proposed schemes and benchmarks with correlated noise (**a**) and with t-distribution noise (**b**).

**Figure 9 entropy-25-01358-f009:**
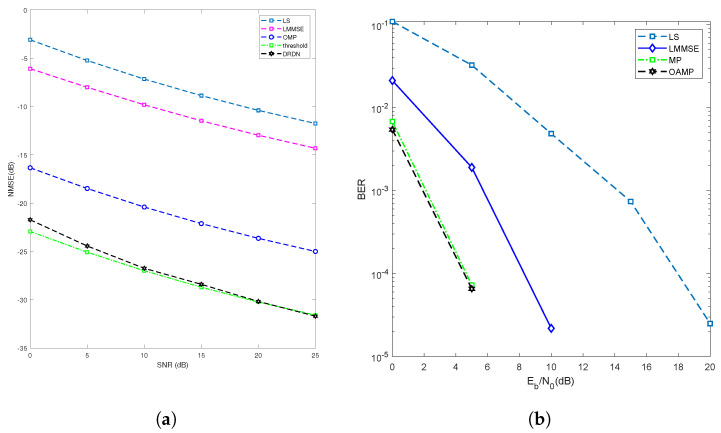
Joint channel estimation and symbol detection (**a**,**b**). (**a**) NMSE comparisons with the proposed schemes and benchmarks with Gaussian noise. (**b**) BER comparisons with the proposed schemes and benchmarks.

**Table 1 entropy-25-01358-t001:** Hyperparameters of the DRDN.

**Input layer:** real-valued matrix with dimension $2k_{\max}\times l_{\max}$		
**Denoising Module:** D denoising blocks share the same construction		
**Layers**	**Operation**	**Filter size**
1	Conv + BN + ReLU	128×(3×2×2)
2~L−1	Conv + BN + ReLU	128×(3×2×128)
3	Conv	2×(3×2×128)
**Subnetwork:** generate the threshold array		
**Module Name**	**Operation**	**Parameters**
$f_{1}\left(\cdot \right)$	Conv + BN + ReLU	32×(3×2×2)
$f_{2}\left(\cdot \right)$	FC + BN + ReLU	2×1×1
**Output layer:** recovery channel matrix of size M×N×2		

## Data Availability

Not applicable.

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
