# Peer review of "A Novel Joint Channel Estimation and Symbol Detection Receiver for Orthogonal Time Frequency Space in Vehicular Networks"

_entropy, 2023, doi:10.3390/e25091358_

Round 1

Reviewer 1 Report

This paper focuses on the theme of integrated sensing and communication (ISAC)-assisted OTFS receiver design, and the estimation of the sensing channel is accomplished through the utilization of a deep residual denoising network (DRDN). Some issues should be addressed as follows.

1.     Authors introduce the background and challenge in nearly half of the abstract, while the contributions and novelty should be better summarized and presented.

2.     The format of abbreviations should be revised. For example, “integrated sensing and communication(ISAC)” should be revised as “integrated sensing and communication (ISAC)”. “equation(9) can be recast as” should be revised as “equation (9) can be recast as”.

3.     Recent advances in the mobile ad hoc networks and efficient communication of vehicular networks should be introduced, such as Impacts of sensing energy and data availability on throughput of energy harvesting cognitive radio networks, IEEE TVT, 2023, DDPG-based joint time and energy management in ambient backscatter-assisted hybrid underlay CRNs, IEEE TCOM, 2023.

4.     Some words are misspelled, such as “Recall thta the steering vectors” in Page 5.

5.     The capitalization of words needs to be revised, such as “To this end, we ultilize On the other hand, based on the received signal, Users estimate the channel matrix, containing sensing information about the surrounding environment.” In page 6.

6.     Authors claim that “a low-complexity OAMP detection technique is employed”, while the complexity analysis is quite simple by only one sentence in Page 11. More details should be provided. Besides, authors also need to demonstrate the low complexity of OAMP.

7.     The parameters in the simulation setups need to be justified. How the values of parameters in simulations are determined?

8.     From simulation results, it seems that more simulations could be conducted, and more competitive schemes could be incorporated in the performance comparison. Besides, authors only introduce the observation from the simulation results, while the corresponding reasons and insights should be provided.

Moderate editing of English language is required for this manuscript.

Author Response

We appreciate your valuable comments very much. In this revision, we 
have carefully revised the manuscript according to your valuable suggestions. We hope this revision can meet your approval. 

Reviewer 2 Report

The authors present a comprehensive approach that integrates sensing and communication (ISAC), utilizing the power of Orthogonal Time Frequency Space (OTFS) to foster efficient channel estimation and symbol detection. This is accomplished through the development of an innovative Deep Residual Denoising Network, expertly crafted for this task.

In terms of estimation accuracy, the proposed neural network design demonstrates remarkable efficiency, outperforming traditional methods. Notably, the symbol detection mechanism, honed by machine learning, significantly excels over conventional systems, demonstrating superior symbol detection performance.

This approach to machine learning-assisted channel estimation and symbol detection offers a fresh perspective in this field. The authors convincingly support the superiority of their methods against existing works through a robust suite of simulation results.

To bolster their claim of superior performance, it may be beneficial for the authors to consider running their simulations in diverse traffic environments. This approach could provide further validation and reinforce the universal applicability of their proposed methods.

Author Response

We thank you for your positive comments. In this revision, we have carefully revised the manuscript based on your valuable comments and suggestions, which has greatly improved the quality of our manuscript. Our hope is this revision meets your approval.

Reviewer 3 Report

The authors nicely represented the represented proposed ISAC system which has better performance and robustness compared to traditional methods.

All the citations are relevant. Still, it is suggested to cite the equation referred to in the article.

During the communication, security plays an important role but authors have not considered it. It is suggested to consider the same

Synchronization is another important concept. Also, the authors have not considered the synchronization methods between RSU and vehicles.

The article may be accepted after the suggested changes.

Author Response

We very much appreciate your positive comments. We have carefully revised the manuscript and justified our assumptions to address your concerns. We hope that our revision would meet your approval.

Reviewer 4 Report

The paper discusses the application of orthogonal time frequency space (OTFS) modulation in vehicular networks to address the challenges posed by high-mobility environments. OTFS modulation transforms time-varying channels into the delay-Doppler domain, enabling high-speed and reliable communication. The authors propose an integrated sensing and communication (ISAC)-assisted OTFS receiver design to perform channel estimation and communication symbol detection. Sensing channel estimation is achieved using a deep residual denoising network (DRDN), while communication symbol detection is performed through orthogonal approximate message passing (OAMP) processing. Numerical results demonstrate the superiority and robustness of the proposed ISAC system compared to traditional methods, offering lower complexity. The authors suggest that this system holds great potential for future wireless communication applications, especially in challenging scenarios with high mobility and interference. After a thorough examination of the manuscript, several comments have been raised as follows:

  1. 1. Definition of Terms and Symbols: It is essential to ensure that all symbols and terms used in the manuscript are well-defined and explained. In Fig. 2, some terms and symbols were not mentioned, and equations for the Heisenberg Transform and Wigner Transform were not included. Additionally, x[k,l] is not defined, and X[n,m] should be in boldface in Fig. 2, among other symbols.

  2.  
  3. 2. Reasonability of Models: Some of the models presented in the manuscript appear to be unreasonable. For instance, the channel gain in Eq. (5) seems to be a constant. It is essential to validate and justify such assumptions or clarify their rationale.

  4.  
  5. 3.Unexplained Operator: The operator denoted by a star in a circle in Eq. (16) is not defined in the manuscript. It is crucial to provide a clear explanation or definition for this operator.

  6.  
  7. 4. Comparison with State-of-the-Art: The manuscript should include a comparison with other state-of-the-art methods to validate the proposed DRDN method's effectiveness fully. Currently, the comparison is limited to some traditional methods like LMMSE and LS.

  8.  
  9. 5. System Model Complexity: The system model appears to be oversimplified. Consider discussing how channel coding or source coding schemes may influence the performance of the DRDN method. Understanding these effects is critical to assessing the practicality and robustness of the proposed approach.

  10.  
  11. 6. Addressing Limitations: The conclusion section should discuss the limitations of the proposed work to provide a more balanced assessment of its contributions and potential areas for future improvement.

  12.  
  13. 7. Inclusion of More References: The manuscript should include additional references to related works and existing literature to support the proposed method's theoretical foundation and positioning within the field.

  14.  
  15. 8. Elaboration on Network Design: More detailed explanations for the network design in Fig. 6 and Fig. 7 are necessary to aid readers in understanding the architecture and implementation of the proposed system.

Addressing these comments would significantly enhance the manuscript's clarity, rigor, and overall contribution to the relevant research area.

Some paragraphs or sentences could benefit from greater conciseness.

Author Response

Thanks for your precious time and effort invested in reviewing this manuscript. Your insightful advice is much appreciated. Please find below our responses to your comments.

Round 2

Reviewer 1 Report

Authors have appropriately addressed my previous comments. The only suggestion is that authors check and revise the format of references.

Reviewer 4 Report

All my previous concerns have been well addressed. The reviewer would like to suggest accepting this work in its current form.